# Lumbo-Pelvic Rhythm Monitoring Using Wearable Technology with Sensory Biofeedback: A Systematic Review

**DOI:** 10.3390/healthcare12070758

**Published:** 2024-03-30

**Authors:** Miguel García-Jaén, Sergio Sebastia-Amat, Gema Sanchis-Soler, Juan Manuel Cortell-Tormo

**Affiliations:** 1Department of General and Specific Didactics, University of Alicante, 03690 San Vicente del Raspeig, Spain; m.garciajaen@ua.es (M.G.-J.); sergio.sebastia@ua.es (S.S.-A.); jm.cortell@ua.es (J.M.C.-T.); 2Health, Physical Activity and Sports Technology (HEALTH-TECH), University of Alicante, 03690 San Vicente del Raspeig, Spain

**Keywords:** feedback, postural balance, low back pain, lumbosacral region, movement control

## Abstract

As an essential lower-back movement pattern, lumbo-pelvic rhythm (LPR) during forward trunk flexion and backward return has been investigated on a large scale. It has been suggested that abnormalities in lumbo-pelvic coordination are related to the risk of developing low back disorders. However, considerable differences in the approaches used to monitor LPR make it challenging to integrate findings from those investigations for future research. Therefore, the aim of this systematic review was to summarize the use of wearable technology for kinematic measurement with sensory biofeedback for LPR monitoring by assessing these technologies’ specific capabilities and biofeedback capacities and exploring their practical viability based on sensor outcomes. The review was developed following the PRISMA guidelines, and the risk of bias was analyzed using the PREDro and STROBE scales. PubMed, Web of Science, Scopus, and IEEEXPLORE databases were searched for relevant studies, initially returning a total of 528 articles. Finally, we included eight articles featuring wearable devices with audio or vibration biofeedback. Differences in protocols and limitations were also observed. This novel study presents a review of wearable tracking devices for LPR motion-mediated biofeedback for the purpose of correcting lower back posture. More research is needed to determine the long-term effectiveness of these devices, as well as their most appropriate corresponding methodologies.

## 1. Introduction

Low back disorders (LBDs) encompass a range of pathologies, such as congenital, developmental, degenerative, traumatic, infectious, inflammatory, and neoplastic conditions, which can lead to pain syndromes, disk degeneration, spondylosis, radiculopathy, stenosis, spondylolisthesis, fractures, tumors, and osteoporosis [1,2,3]. Given their multifactorial nature and the complexity and variations that exist in therapeutic approaches, LBDs constitute a significant global health concern [1,2,3]. LBDs are a leading cause of disability; each year, they affect the wellbeing of the general population and place an economic burden on healthcare systems worldwide [4]. Worldwide, the direct and indirect costs of treating LBDs are estimated to exceed USD 100 billion per year, and the number of people with LBDs is forecast to increase in the coming decades due to muscular deconditioning caused by the physical inactivity and sedentary lifestyles of modern societies [4,5]. Therefore, effective countermeasures and therapeutic interventions are urgently needed in order to address this growing health problem [6].

Due to the multifactorial nature of LBDs and the complexities involved in their treatment, the management of LBDs requires a combination of active therapies such as exercise, education, and prevention and passive therapies such as manual therapies, assistive devices, and medications [6,7,8]. In fact, the most widely accepted strategies for reducing the incidence of LBDs are therapeutic spinal exercises and continuous lumbo-pelvic control and postural monitoring [9,10]. Accordingly, maintaining the stability and alignment of the lumbo-pelvic complex during functional multiplanar movements and avoiding end-range bending positions during lumbo-pelvic motion are crucial considerations within specific motor control training; they have exhibited positive effects in the prevention and management of LBDs, as suggested by existing research [9,10,11,12].

The motional pattern of the trunk in the sagittal plane is the result of the joint action of the lumbar spine and the pelvis. The contribution of the flexion–extension movements of the lumbar spine and the tilting movements of the pelvis to this synchronized motion is referred to in the literature as lumbo-pelvic rhythm (LPR) [12,13,14,15]. Extensive research has focused on LPR to understand the neuromuscular control of the lower back region, define each segment’s contribution to lower back motion, and discriminate between people with low back pain (LBP) and asymptomatic individuals [12,13,14,16,17,18,19]. In this context, biomechanical abnormalities in LPR may indirectly indicate alterations in the neuromuscular coordination of the lumbo-pelvic region and shifts in load distribution within lower back tissues, both of which are key factors contributing to the development of LBD [11,12,16,20,21]. Thus, the precise assessment of LPR is a crucial precursor to the implementation of effective prevention and rehabilitation programs [12,17,22,23].

The assessment of LPR may involve evaluating the relative contributions of lumbar and pelvic sagittal rotation to the endpoints of lower back motion; alternatively, it may entail a more comprehensive kinematic assessment of the timing and magnitude of these contributions within the entire range of forward flexion and backward extension [17,18,24,25]. Researchers and clinicians have employed indirect in vivo methods, such as lumbo-pelvic kinematics and the electromyography of the trunk musculature, in assessing the mechanical characteristics of the lower back; such methods evaluate patient conditions and guide management decisions [25,26,27,28,29,30,31,32,33,34,35,36]. However, studies utilizing these indirect measures, especially kinematic measurements, have exhibited significant variability in their outcomes and lack conclusive findings [13,37].

Kinematic measurements and the monitoring of LPR and lower back posture are commonly used in clinical settings during patient exams and physical therapy assessments [12,13,17,31,33,34]. However, traditional laboratory or clinical methods—such as radiographic assessment and the use of goniometers or photogrammetric and optoelectrical systems—have been criticized for being expensive and impractical; they also demonstrate limitations in terms of their objective and continuous measurements of low back posture and real-time biofeedback [38,39]. Earlier studies have used optical motion capture systems [39,40,41] and electromagnetic tracking devices [42,43,44] to accurately assess the lumbar spine and LPR, but these techniques tend to be cumbersome and limited to the laboratory; as a consequence, they are not suitable for widespread use in clinical, rehabilitative, or preventive exercise settings [45,46,47,48].

One promising solution to these limitations is the use of wearable measurement technology, which has attracted considerable scientific interest due to its potential to provide real-time insights into the kinematic aspects of human movement through continuous, dynamic, and minimally invasive monitoring [49,50,51,52,53]. This technology has demonstrated the ability to monitor the posture of the lower back and LPR, providing immediate biofeedback in the event of incorrect lumbo-pelvic movements or persistent poor posture [32,50,52]. Among wearable sensing approaches, inertial measurement unit (IMU)-based systems have gained prominence in monitoring lumbo-pelvic motion [32,54,55]. Tracking devices such as Xsens, Vimove, Lumbatex, Ergotex, Spineangel, PostureCoach, Valedomotion, and BodyGuard have been validated and employed for monitoring back movements in the laboratory as well as in rehabilitative and clinical settings [35,56,57,58,59,60,61,62,63,64,65]. These inertial tracking devices are non-invasive, low-cost, compact, and portable; they comprise a triaxial accelerometer, gyroscope, and magnetometer unit, all of which have been proven to be valid and accurate tools for the quantification and management of three-dimensional spinal and pelvic motion in laboratory, clinical, and rehabilitative settings [32,49,54,55]. In the area of LPR monitoring, inertial sensors have become indispensable tracking devices for patient assessment and the monitoring of low back rehabilitation exercises in clinical settings [13,14,34].

Wearable sensor systems for postural monitoring have demonstrated the potential to provide real-time biofeedback in order to promote individuals’ self-correction of improper LPR and persistently poor lower back posture [32,50,63,66]. Feedback refers to afferent sensory information resulting from the movement of the human body in relation to the environment [67]. It can be intrinsic (e.g., proprioception), when the brain perceives somatic information from body movement, or extrinsic (e.g., sensory biofeedback) when provided by an external source [13,67]. Feedback may be provided directly by the health professional or indirectly via simple interfacing (e.g., mirror, video) or sensory biofeedback (e.g., IMU). This may be in real time or delayed, and different sensory modalities (e.g., auditory, visual, haptic) and frequencies or levels (e.g., continuous, instantaneous, intermittent) can be employed [59,62,63,67,68]. The integration of sensory biofeedback into wearable technology may act to mitigate sensorimotor disturbances and enhance intrinsic feedback, assisting individuals to correct impaired LPR and improve their lumbo-pelvic motor control [35,63,67]. Given its potential to control LPR and lower back posture, wearable technology has been hailed as a feasible option for implementation in both clinical and real-world settings for daily use [12,13,52,66].

Nevertheless, given the variety of wearable tracking devices, biofeedback options, and timescales available for lumbo-pelvic monitoring, it is essential to optimize biofeedback produced by wearable sensors. As persistent poor LPR posture is associated with the development and exacerbation of several LBDs, this optimization of biofeedback is required to enable users to adjust motor strategies for real-time LPR corrections [66,69]. Furthermore, we must efficiently integrate previous research related to LPR monitoring and appraise the currently available approaches to controlling LPR and monitoring lower back posture using wearable sensor systems [13,14,33].

Therefore, the aim of this review was to summarize the use of wearable technology for kinematic measurement with sensory biofeedback for LPR monitoring, with the goals of (1) assessing the capabilities of current wearable tracking devices for measuring lumbo-pelvic motion and posture, (2) identifying studies that have implemented such devices with sensory biofeedback, and (3) exploring the practical feasibility of integrating these devices based on sensor outcomes. The aim of this systematic review is to synthesize data from different scientific fields (engineering, physiotherapy, rehabilitative medicine, etc.) that use wearable tracking devices in order to improve LPR control and lower back posture and function.

## 2. Materials and Methods

This systematic review was prepared and carried out following the PRISMA guidelines (Preferred Reporting Items for Systematic Reviews and Meta-Analyses) (see Appendix A) [70,71].

### 2.1. Literature Search Strategy

IEEEXPLORE, Web of Science (all databases), PubMed, and Scopus databases were used for the various searches. Initially, our search was limited to articles written in English and Spanish and published between 2004 and 2024.

Our search utilized MeSH and non-MeSH terms, and its strategy is presented in Table 1. Additionally, following the latest PRISMA update [70], the snowball technique was used to search for articles included in existing relevant reviews. These articles were identified within a reference list of studies or within citations of studies. Searches of websites considered eligible in terms of both organization and date were also conducted [70,72].

### 2.2. Study Selection

The studies for this systematic review were selected independently by three reviewers. For this purpose, the Rayyan directory was used, following previously established inclusion and exclusion criteria.

After a preliminary consensus on the selected studies was reached, said studies were included according to the following inclusion and exclusion criteria. Firstly, to be considered eligible, studies had to present a study sample consisting of individuals with or without LBDs, including DL. Secondly, the studies had to provide kinematic measurements of both the lumbar spine and the pelvis (LPR), in addition to providing sensory biofeedback. These kinematic measurements needed to have been performed using portable monitoring devices.

On the other hand, studies with a study sample age below 18 years, studies conducted on animals, studies not focused on a kinematic analysis of LPR movement or lumbo-pelvic posture, studies not using portable monitoring devices, and studies written in a language other than English or Spanish were excluded. Finally, all studies that were not clinical trials, randomized controlled trials, case reports, preliminary studies, or descriptive studies were also excluded.

Figure 1 shows the planned flowchart of the systematic review proposed within this protocol.

### 2.3. Data Collection

After selecting the studies, two researchers carried out the data extraction process independently. For each investigation, the names of the authors, year of publication, population and experimental/control group characteristics, device used and its exact location, reported biofeedback, trigger procedure, and sensor outcomes were included in the summary table.

### 2.4. Quality Assessment of the Reviewed Articles

The quality of the experimental studies included was assessed utilizing the Physiotherapy Evidence Database (PEDro), which evaluates the methodological quality, including internal validity and statistical information, of randomized clinical trials (RCTs) and other experimental trials; it is considered a valid measure of both internal validity and the comprehensiveness of reporting [73]. Except for the first item, each element of the scale contributed one point to the total PEDro score, which ranges from 0 to 10. Based on the cumulative score derived from these criteria, studies scoring 6 or higher were considered to be of high quality, 4–5 indicated moderate quality, and less than 4 suggested low quality [74].

The quality of the observational studies included in this review was evaluated using an adapted version of the STROBE (STrengthening the Reporting of OBservational studies in Epidemiology) checklist, following the example of existing relevant research [75,76]. This scale comprises 10 specific items selected from the original checklist (items 1, 3, 6, 8, 11, 14, 18, 19, 20, and 22), with each item scored on a binary scale ranging from 0 (not fulfilled) to 1 (fulfilled). Based on the cumulative score derived from these STROBE criteria, the articles were categorized as low-quality (scoring ≤ 7 points) or high-quality (scoring > 8 points).

Two researchers conducted the methodological evaluation process. Discrepancies between the reviewers’ evaluations were resolved with a consensus, ensuring the integrity and consistency of the evaluation process (Table 2 and Table 3).

## 3. Results

The search initially identified 528 studies. After applying the inclusion and exclusion criteria, six investigations were finally included for analysis, along with two other studies identified by the snowball search technique. This meant a final number of eight studies included in this systematic review, as shown in Table 4.

### 3.1. Quality of Reviewed Articles

The methodological quality and risk of bias of the selected studies were assessed using the PEDro and STROBE checklists, as shown in Table 2 and Table 3, respectively. Among the eight included studies, five were randomized controlled trials (RCTs) [59,60,61,62,65], one was a non-randomized clinical controlled trial [64], one was an RCT pilot study [63], and one was a descriptive study [47].

The scores on the PEDro scale ranged from five to nine points (out of ten), with a median of eight points. Specifically, one study met nine criteria [60], three studies met eight criteria [59,62,63], one study met seven criteria [61], and two studies met five criteria [64,65], indicating a low risk of bias. Based on the cumulative score derived from the PEDro criteria, five studies scored 6 points or higher and were considered of high quality, accounting for 71.43% of the total studies included in the analysis, while two scored 5 points, indicating moderate quality (28.57 of the total studies). It is notable that none of the articles received a score less than four, suggesting low quality according to the PEDro criteria [73,74].

However, the descriptive study [47] received a score of 6 from the two researchers, indicating low quality according to the STROBE criteria [75,76]. The PEDro assessments conducted by the two researchers were consistent, yielding mean review scores of 7.00 ± 1.22 and 7.5 ± 1.32, respectively. These evaluations were carried out independently, and the reviewers were blinded to each other’s scores, thereby ensuring impartiality and minimizing potential biases. Finally, discrepancies were resolved with a consensus, resulting in a final mean score of 7.14 ± 1.46 for the PEDro assessments.

### 3.2. Descriptive Analysis of the Studies

A sample totaling 464 subjects was analyzed. In general, all studies included a population over 18 years old, with a mean age ranging between 23.6 ± 3.1 and 49.6 ± 12.4 years. Among the pathologies suffered by the study population, lumbar pain stands out, and was mostly non-specific, chronic, or acute. However, participants without lumbar pain were included in five out of the eight studies analyzed. All the studies analyzed used IMU-based technology, utilizing two or three sensors for kinematic measurements. In addition to kinematic movement, all studies provided sensory biofeedback. Five of the reviewed studies used real-time auditory biofeedback, one study used real-time visual feedback, another used haptic vibration biofeedback, and the last study combined visual and haptic vibration real-time biofeedback. Although the eight studies provided information on lumbo-pelvic kinematics, we observed some methodological differences in relation to the locations and attachments of these IMU sensors.

The work of Ribeiro et al. [59,60] involved the Spineangel, which is composed of a single monitoring device positioned on the hip of the subject being evaluated. The remaining six studies involved IMU sensors located in the thoracolumbar and sacral areas (between the T1–S2 interval), in some cases including a third sensor on the lateral femoral condyle [62]. In one study [65], a specific lumbo-pelvic location between L3 and S2 was chosen for IMU sensor placement.

The devices used for kinematic measurements of LPR motion and lower back posture were Spineangel, PostureCoach, Valedo^®^motion, BodyGuard™, and Vimove. All of these wearable tracking devices have been used to provide short-term biofeedback to individuals with back pain or asymptomatic individuals in workplace settings (for example, among caregivers) as a means of preventing injury [61,64]. On the other hand, Kent et al. [63] used their device, Vimove, as a biofeedback training device to control individuals’ fear of performing certain movements. Finally, only one of the analyzed articles utilized their device, Spineangel, to monitor whether workers exceeded a given postural threshold via the device’s audio feedback [60].

However, despite the promising results achieved by these wearable tracking devices, the aforementioned articles highlight certain limitations regarding the IMU sensor biofeedback provided by these devices. Specifically, Ribeiro et al. [59], who used Spineangel as a lumbo-pelvic postural monitoring device to measure exposure to postural movements, highlighted its main limitations: a lack of adherence and potentially limited validity in the precise degree-scale measurement of movement. Other studies indicated a need to use more realistic measurement environments and a more accurate fixation of sensors [61].

Finally, the reviewed studies suggested that more follow-up studies will be needed to monitor the retention of sensor-based feedback, the measurement of areas whose activation could occur as compensatory activity, the inclusion of measurements in various planes of movement, and study samples that are appropriate for identifying differences between constant extrinsic feedback and a control group [59,62,64].

## 4. Discussion

The use of wearable technology in tracking and monitoring human movement is growing in prominence across various disciplines, including biomechanics and kinesiology and rehabilitative medicine and physiotherapy [50,51,66,69]. These technologies’ growing adoption in clinical and therapeutic settings is due to their capacity to provide real-time biofeedback, enabling individuals to correct improper LPR and chronic lower back posture issues and thus serving as valuable tools for interventions in LBDs [50,52,53,69]. In this area of research, most studies published in the past decade have primarily focused on monitoring the lumbar spine [35,39,43,55,68]. However, the sagittal motion of the lower back, known as LPR, results from coordinated movements between the lumbar spine and the pelvis [13,14,34]. Due to this motion, patterns in LPR have been studied in depth and monitored independently of lumbar spine motion; however, we must also carry out more comprehensive reviews of LPR and its importance in the monitoring of lower back motion using biofeedback mediated by wearable technology [13,17,33]. Hence, this systematic review intends to synthesize data on the utilization of wearable tracking devices for enhancing individuals’ control of their LPR and improving lower back posture and function. To do so, we analyzed the capabilities of currently available wearable tracking devices in measuring lumbo-pelvic motion and posture, identifying studies that have implemented these devices with sensory biofeedback and exploring sensory outcomes to assess the practical feasibility of integrating these wearables into therapeutic and clinical settings.

### 4.1. Wearable Technology for Lumbo-Pelvic Rhythm Monitoring

This is a novel systematic review of eight studies providing an overview of wearable tracking devices specifically utilized for LPR monitoring; it offers insights into the effects of biofeedback employed during controlled, simulated, and real tasks in various settings. Our findings indicate that despite the wide range of potential devices available for use (i.e., strain gauges, flex or electronic sensors, fiber-optic goniometers, inductive sensors, etc.), the most commonly utilized tool for LPR monitoring was the IMU sensor [47,59,60,61,62,63,64,65]. Even though the study selection criteria for portable monitoring devices within this review were not limited to IMU-based systems, the findings revealed that each of the reviewed studies exclusively used IMU sensors to comprehensively monitor the lumbo-pelvic region. This finding underscores that at present, the use of these wearable devices has become indispensable in monitoring the lower back, as also presented in other relevant studies [50,66]. However, sensor outcomes varied among studies, including different recordings of lumbo-pelvic spine sagittal motion [61,64], the monitoring of static posture for estimating spinal curvature [47], and controlling lower back posture through real-time biofeedback to enhance sensorimotor control and function, thereby reducing symptoms associated with LBDs in controlled and real-world settings (e.g., through tracking the frequency with which specific postural thresholds were surpassed) [59,60,62,65].

The differences observed between outcomes in controlled and real-world environments (e.g., workplaces and clinical and therapeutic settings) could explain the discrepancies in sensor usage and outcomes obtained. Finally, based on the favorable conclusions reported within the reviewed studies, it seems that tracking devices such as Spineangel, PostureCoach, Valedo^®^motion, BodyGuard™, and Vimove may hold clinical significance. The utilization of extrinsic sensory biofeedback appears beneficial in aiding individuals with and without LBDs to improve their LPR sensorimotor control and preserve their lower back posture. It may contribute to the prevention of postures and LPR movements that might trigger lower back pain or exacerbate its symptoms, particularly in cases wherein an LBD already exists. However, given the limited number of available studies on the topic of this review, further research conducted in clinical and therapeutic settings and focusing on lumbo-pelvic motion control is necessary; new investigations should utilize both existing devices and newly proposed wearable sensors. Such studies will be essential to establish a more comprehensive body of evidence regarding the efficacy of wearable sensory biofeedback in monitoring LPR for the prevention and rehabilitation of LBD. Additionally, the lack of studies involving long-term monitoring and control of LPR using real-time biofeedback limits our ability to draw conclusions regarding the immediate and short-term effects of the biofeedback provided. Therefore, further research is needed to develop devices capable of monitoring for longer periods (e.g., 24–48 h) in order to explore the extended effects of wearable-technology-mediated sensory biofeedback on individuals’ neuromuscular control of their lower back.

On the other hand, the characterization of LPR varied across the reviewed studies depending on the device used. In all cases, LPR was taken to refer to the relative motion between the lumbar and pelvic segments, with reference to either local (e.g., thigh) or global (e.g., gravity direction) axes and utilizing two or three sensors accordingly. Most studies employed two sensors to monitor LPR, positioning IMUs at the end of the thoracolumbar area and the pelvis, with the gravity vector serving as a global axis of reference [47,59,60,61,63,64,65]. A single study [62] integrated a third sensor on the lateral femoral condyle as a local reference, facilitating subsequent kinematic measurements of the lumbo-pelvic area. This variability in characterization may impact sensors’ outcomes, particularly when discerning the distinct contribution of the lower extremity to LPR. As proposed by Vazirian et al. [13], the global approach to measuring lumbo-pelvic motion takes into account the involvement of all lower extremity joints in LPR, whereas the local approach concentrates solely on the influence of hip joint motion on LPR. Consequently, the measured motion may vary depending on the chosen measurement methodology regarding the location and number of sensors used [12,13].

Finally, it is important to consider that kinematic measurements of LPR motion using IMU sensory biofeedback require the attachment of the sensors to previously determined anatomical landmarks within the lumbo-pelvic segment [13,77]. Thus, the movements of two or more anatomical landmarks should be tracked (i.e., making a line or a plane) when using IMU sensors [13,77]. The location of these anatomical landmarks—used to place the pelvic sensor for measuring LPR motion—in the reviewed studies included L5, S1, and S2 [47,62,65]. On the other hand, the location of the anatomical landmark for thoracic sensors that measured LPR motion included the lower segments of the thoracolumbar joint: L1, T12, and T10 [47,61,62,64]. Alternatively, the work of Ribeiro et al. [59,60] employed the Spineangel, which is composed of a single monitoring device positioned on the hip of the evaluated subject. Based on variability in the timing and magnitude of the lumbar spine’s contribution to the forward bending of the trunk during LPR motion, it is conceivable that the effectiveness and accuracy of the Spineangel as a lumbo-pelvic postural monitoring device providing real-time biofeedback could be compromised when the forward inclination threshold is reached, given the unique sensor location on the hip [13,33].

### 4.2. Characteristics of Sensory Biofeedback

The data extracted in this review provide insights into the main types and triggers of biofeedback reported by the selected studies detailing wearable devices for controlling lumbar and pelvic movements. First, our findings showed that the most common real-time biofeedback used was audio alerts (featured in five out of eight studies) [47,59,60,61,64], with one of these studies employing audio feedback plus vibration [63]. The other two devices employed haptic [65] or visual [62] real-time biofeedback. Audio biofeedback, according to the analyzed articles, can be differentiated into two forms: intermittent and continuous. Continuous biofeedback appears to be more beneficial than intermittent biofeedback, as suggested by the outcomes of the studies analyzed. While intermittent audio biofeedback may promote optimal adherence, several analyzed studies revealed a drawback: this type of biofeedback only provides information when the person exceeds a predetermined threshold, which is sometimes insufficient to control lumbo-pelvic motion [59,60]. It appears that offering continuous feedback after training tasks could enhance the success of these interventions [64]. In fact, continuous or gradual audio feedback could be a good method of inducing changes in postural behavior and reducing the risk of back injury [68]. Furthermore, it is noteworthy that two studies utilized continuous visual and composite biofeedback (i.e., both audio and vibration) [62,63]. Both works concluded that the reported information could be beneficial and useful for the treatment of people with LBDs [62,63]. However, while the findings generally support the capacity of audio, vibration, or visual biofeedback (whether continuous, gradually continuous, or intermittent), to expand the existing body of evidence on the use of biofeedback for LPR control, additional studies with larger sample sizes and longer follow-up periods will be needed. Additionally, as suggested by O’Sullivan et al. [65], it is crucial to acknowledge the challenge of maintaining the participants’ blindness to the nature of the feedback administered in order to ensure the accuracy of sensor outcomes.

Finally, after reviewing the eight studies included in this analysis of biofeedback usage, there were certain notable limitations, as reported by the authors. Firstly, determining the optimal timing for integrating this type of biofeedback training and conducting long-term follow-up evaluations represent advisable next steps [62,64]. Secondly, studies have emphasized some concerns about sample selection bias and the participation of certain industries, the partial blinding of clinicians, and limitations related to the measuring range of motion and participant characteristics (which should occasionally be adjusted to more realistically represent real-world people and circumstances) [59,61,63]. Lastly, some authors have suggested the introduction of a greater number of sensors for monitoring adjacent joints that could interfere with lumbo-pelvic motion, as well as better sensor attachment to the skin [61,62].

### 4.3. Practical Applications

In this section, we critically evaluate the practical application of this technology, highlighting its potential applicability in different fields such as clinical and therapeutic settings.

The accuracy of portable devices for monitoring LPR may vary depending on the sensor used, the placement of the sensor, and the activity performed [12,13,66]. It is crucial to assess the validity and reliability of these devices in different clinical settings and patient populations [12,49,67]. Although sensory biofeedback technology offers potential benefits, its effective integration into clinical settings requires logistical considerations, such as staff training, interoperability with existing data-recording systems, and the availability of financial resources [12,66,67].

In rehabilitative settings, wearable devices could play a pivotal role in ensuring the correct execution of rehabilitative exercises by monitoring progress remotely and providing immediate feedback on posture [49,51]. Regarding the latter, the capacity of wearable systems to provide real-time feedback could empower users to promptly adjust their posture and movement, potentially enhancing biomechanics and mitigating strain on musculoskeletal structures [53,68]. This technology could also foster a heightened sense of body awareness, facilitating the adoption of healthier movement patterns and postures [50,66]. Such heightened awareness may prove particularly beneficial for individuals grappling with postural abnormalities or lumbo-pelvic injuries, enabling them to actively incorporate compensatory behaviors into their daily activities [26,60,63,78].

Furthermore, wearable devices measuring spinal posture have potential applications in the prevention, monitoring, and treatment of LBD. By continually monitoring posture, these devices aid in maintaining proper alignment, reducing the risk of musculoskeletal issues, and supporting the long-term management of chronic conditions [51,67]. This multifaceted approach could contribute to muscle strengthening, postural correction, and the overall efficacy of rehabilitation programs [28,30,53,62,65]. Additionally, the longitudinal tracking of LPR data could allow for personalized assessments of patient progress, facilitating the tailored adaptation of rehabilitation interventions to individualized needs [12,13]. Beyond traditional rehabilitation settings, the utilization of wearable systems for posture monitoring may assist in tele-rehabilitation endeavors by offering remote rehabilitation services [51,67,79]. This remote approach is especially beneficial for individuals who are far from medical services or who have limited mobility, such as the elderly or people with reduced mobility [2,80]. Finally, the data provided by wearable sensors could serve as a valuable tool in clinical decision-making processes. Leveraging this wealth of information enables healthcare professionals to make informed choices regarding patient care and treatment strategies [51,67].

### 4.4. Limitations and Future Research

Upon completion of this review, significant gaps in the literature emerged. It is clear that studies involving long-term monitoring are lacking; this deficiency hampers our comprehension of how posture evolves, over time and in different situations, as a consequence of wearing sensors with biofeedback. Additionally, the placement of wearable devices and the controlled environments in which studies are often conducted can disrupt the natural execution of movements and potentially influence results. These circumstances contribute to a lack of understanding of the capacity and effectiveness of these devices over prolonged monitoring periods, particularly in uncontrolled real environments.

One of the reasons for the limited number of studies included in this review is the predominant focus of existing research on the lower back area; the broader lumbo-pelvic region is often disregarded. While understanding that the lower back is important, considering the pelvis in posture and movement assessments is equally so. In this way, we might gain a more comprehensive insight into human biomechanics. Therefore, future research efforts should aim to address these gaps in order to effectively integrate these devices into clinical and rehabilitation fields.

## 5. Conclusions

This systematic review is, to the best of our knowledge, the first to carry out a focused assessment of the wearable tracking devices with sensory biofeedback that are currently employed for LPR monitoring and lower back control. It encompasses studies on LPR motion and summarizes their methods of kinematic measurement, offering insights into the effects of biofeedback utilized during controlled, simulated, or real tasks performed in various settings (e.g., work, laboratory, and rehabilitation).

In summary, our results show that kinematic measurements have primarily been conducted using IMU sensors; they also indicate that tracking devices such as Spineangel, PostureCoach, Valedo^®^motion, BodyGuard™, and Vimove may hold clinical significance. However, across these studies, there was variability in the anatomical landmarks utilized to measure lumbar and pelvic motions, which might have affected the outcomes of these devices’ sensors. Real-time audio alerts were the most commonly used form of biofeedback, and were administered intermittently and continuously. The main purpose of these devices was to control LPR using sensory biofeedback for the purpose of correcting lower back posture; the clinical utility of postural sensors can be optimized and their data leveraged to enhance clinical decision making, thus potentially influencing verdicts on the necessity of surgical intervention.

Finally, this review uncovers limitations within existing studies, which require attention. However, the literature available for review was somewhat limited. Consequently, further research concentrating on methodology, precision, and long-term outcomes is needed, so that we might enhance our understanding of the clinical relevance of these devices. Additionally, investigations into practical enhancements, such as sensor transparency and the practical forms of providing biofeedback, should be conducted before considering commercialization and widespread adoption.

## Figures and Tables

**Figure 1 healthcare-12-00758-f001:**
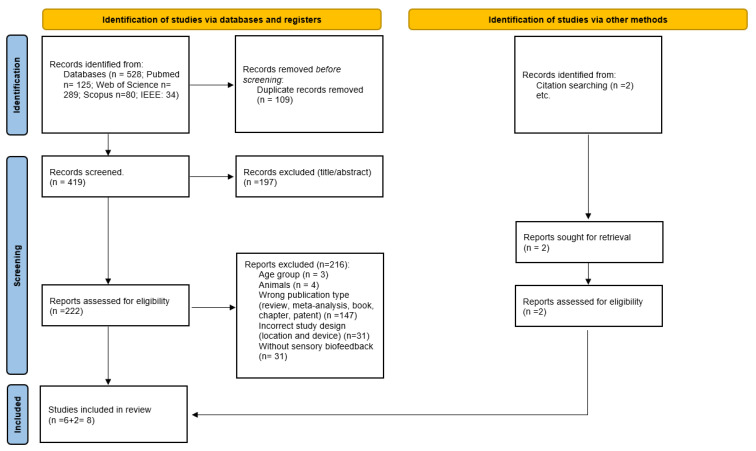
Flowchart of this systematic review’s search strategy.

**Table 1 healthcare-12-00758-t001:** Search strategy.

Area	Search Strategy
Sensors	(sensors OR device OR system OR wearable OR portable OR accelerometer OR inertial sensor OR gyroscopes OR goniometer) AND
Biofeedback	(feedback OR biofeedback) AND
Purpose	(monitoring OR postural control OR ergonomics OR postural health OR stabilization) AND
Outcomes	(motion OR position OR movement OR posture) AND
Location	(lumbar curvature OR lumbar spine OR lumbopelvic rhythm OR lumbo-pelvic rhythm OR spine OR lumbopelvic control OR sacroiliac joint)

**Table 2 healthcare-12-00758-t002:** Methodological quality assessment and risk of bias (conducted using PEDro criteria).

Study	PEDro Criteria											
	Items											Total
	1 *	2	3	4	5	6	7	8	9	10	11	
Ribeiro et al., 2014 [59]	1	1	1	1	1	0	0	1	1	1	1	8
Owlia et al., 2020 [61]	1	1	1	1	0	0	0	1	1	1	1	7
Matheve et al., 2018 [62]	1	1	1	1	1	0	0	1	1	1	1	8
Kent et al., 2015 [63]	1	0	1	1	1	0	1	1	1	1	1	8
Kamachi et al., 2021 [64]	1	0	0	1	0	0	0	1	1	1	1	5
Ribeiro et al., 2020 [60]	1	1	1	1	1	1	1	1	0	1	1	9
O’Sullivan et al., 2013 [65]	1	1	0	1	0	0	0	1	1	1	1	5

Item legend (PEDro scale): * = was not counted for the final score; 1 = one point allocated; 0 = no points allocated. 1. Eligibility criteria were specified. 2. Subjects were randomly allocated to groups (in a crossover study, subjects were randomly allocated an order in which treatments were received). 3. Allocation was concealed. 4. The groups were similar at baseline regarding the most important prognostic indicators. 5. There was blinding of all subjects. 6. There was blinding of all therapists/researchers who administered the therapy/protocol. 7. There was blinding of all assessors who measured at least one key outcome. 8. Measures of at least one key outcome were obtained from more than 85% of the subjects initially allocated to groups. 9. All subjects for whom outcome measures were available received the treatment or control condition as allocated or, where this was not the case, data for at least one key outcome were analyzed by “intention to treat”. 10. The results of between-group statistical comparisons were reported for at least one key outcome. 11. The study provided both point measures and measures of variability for at least one key outcome.

**Table 3 healthcare-12-00758-t003:** Methodological quality assessment and risk of bias (conducted using STROBE criteria).

Study	STROBE Criteria										
	Items										Total
	1	2	3	4	5	6	7	8	9	10	
Wong et al., 2008 [47]	1	1	0	1	1	0	1	0	1	0	6

Item legend (STROBE scale): 1. Provide in the abstract an informative and balanced summary of what was undertaken and what was found. 2. State specific objectives, including any prespecified hypotheses. 3. Provide the eligibility criteria, and the sources and methods of selection of participants. 4. For each variable of interest, give sources of data and details of methods of assessment (measurement). Describe the comparability of assessment methods if there is more than one group. 5. Explain how quantitative variables were handled in the analyses. If applicable, describe which groupings were chosen and why. 6. Provide the characteristics of the study’s participants (e.g., demographic, clinical, social) and information on exposures and potential confounders. 7. Summarize key results with reference to the study’s objectives. 8. Discuss the limitations of the study, considering sources of potential bias or imprecision. Discuss both the direction and magnitude of any potential bias. 9. Present a cautious overall interpretation of results, considering objectives, limitations, multiplicity of analyses, results from similar studies, and other relevant evidence. 10. Disclose the source of funding and the role of the funders in the present study and, if applicable, in the original study on which the present article is based.

**Table 4 healthcare-12-00758-t004:** Summary of encompassed research and findings.

	Main Aim	Participants	Health Condition	Device and Location	Biofeedback Type	Biofeedback Trigger	Procedure	Sensor Outcomes
Ribeiro et al., 2014 [59]	To evaluate the feasibility of a trial investigating the effectiveness of a lumbo-pelvic monitor as a feedback device for modifying postural behavior during daily work-related activities.	*n* = 62 workers (age: 49.6 ± 12.4) (CG: 16; IFG: 23; CFG:14) worked a minimum of 20 h/w	With or without symptoms of non-specific low back pain	D: Spineangel L: Cadera	Auditory Intermittent Feedback	Feedback occurred when three signals (ROM, frequency, and duration) were activated. ROM: 45° lumbo-pelvic inclination forward. Frequency: when more than twice this threshold is exceeded for one minute. Duration: when inclination was maintained for 5 s.	Participants wore the device for 6 weeks (working days). The number of times workers exceeded the accumulated postural threshold per hour was counted.	The frequency of exceeding the postural threshold was recorded.
Owlia et al., 2020 [61]	To investigate the effect of real-time biofeedback on time spent by caregivers in end-range lumbar spine flexion.	*n* = 20 informal caregivers. CG (*n* = 10; age= 24.7 ± 2.7), IG (*n* = 10; age = 28.1 ± 6.4)	Healthy	D: PostureCoach L: upper sensor in T10; lower sensor near the sacrum	Auditory Continuous Feedback	Continuous signal when the spine flexion angle exceeded a predefined threshold (70% maximum flexion). From 0 to 20° before 70% it is not activated; from 20° to 70% the signal is intermittent and ≥70% continuous.	In the laboratory, a simulation of a house was carried out, where the participants had to perform different tasks with an actor who pretended to be an elderly person. The GI was trained through a video.	Spine flexion was recorded.
Matheve et al., 2018 [62]	(1) To evaluate whether sensor-based feedback is more effective in improving lumbo-pelvic movement control compared to feedback from a mirror or no feedback in patients with CLBP; (2) To evaluate whether patients with CLBP are as able as healthy people to improve lumbo-pelvic movement control.	*n* = 91; age: 37.8 ± 3.8 Sensor group (SG) Mirror group (MG) CG (no feedback)	Chronic non-specific LBP *n* = 44 (>3 months, ≥3 days/week) and Healthy *n* = 47	D: Valedo^®^ motion L: Placed on L1 and S1 and 20 cm above the lateral femoral condyle	Visual Continuous Feedback	SG: via an avatar on a computer screen in front of the participants. MG: A large mirror was placed laterally to the participants.	Participants completed a lifting task followed by a waiter’s bow. Each task was performed five times at a self-selected speed. Waiter’s bow: three sets of six repetitions with one minute of rest and the assigned feedback in each case.	The effectiveness of sensor-based feedback compared to mirror feedback and no feedback in improving lumbo-pelvic movement control performance was assessed.
Kent et al.,2015 [63]	(1) To test whether changing painful lumbo-pelvic movement patterns using motion sensor biofeedback in people with low back pain could lead to reduced pain and limitations on activity compared to guideline-based care; (2) To facilitate sample size calculations for a fully powered test.	*n* = 112 adults with lumbar pain movement: biofeedback group = 58 (age 39 ± 12); guidelines-based care group = 54 (age 48 ± 12)	Lumbar pain (or back-related leg pain) intensity ≥3 (0–10) duration subacute (3–12 weeks) or chronic (≥12 weeks)	D: ViMove L: On toraco lumbar junction and on the upper sacrum	Auditory and Vibration Feedback	The specialist determined the threshold from which biofeedback would be activated for each patient. This threshold was determined from a previous analysis in which the positions that cause pain were determined.	All participants in both groups were assessed at baseline and attended a total of 6 (sub-acute episode duration patients) to 8 (chronic episode duration patients) consultations over a 10-week treatment period. All participants wore the ViMove motion sensor system for 4 to 10 h in their activities of daily living, during and after each treatment session (6 to 8 times) over the 10-week treatment period.	Self-reported pain intensity (VAS) and activity limitation (Roland–Morris Disability Questionnaire).
Kamachi et al.,2021 [64]	To evaluate the effectiveness of a two-day training intervention (including PostureCoach and an educational video) for its ability to decrease the amount of time spent in extreme spinal flexion.	*n* = 20 healthy participants Age CG = 23.6 ± 3.1 Age IG = 24.4 ± 3.7	No back pain in the last six months, or any musculoskeletal disorder related to the spine.	D: PostureCoach L: upper sensor in T10; lower sensor near the sacrum	Auditory Continuous Feedback	Researchers recorded when the spinal flexion angle exceeded a predefined threshold (70% of maximum flexion). From 0 to 20° before 70% it was not activated, and from 20° to 70% the signal was intermittent and ≥70% continuous.	Participants were asked to complete a series of simulated care tasks with a “patient” actor in a bedroom, on a couch, and in a bathroom.	Lumbar flexion was recorded.
Wong et al., 2008 [47]	Introduced triaxial accelerometers and gyroscopes to detect postural changes in terms of variation in spinal curvature in the sagittal and coronal planes and to demonstrate the performance of the posture monitoring system during daily activities.	*N* = 9; age: 25.2 ± 4.8	----	D: three inertial sensor modules of the detection system. L: T1/T2, T12 and S1	Auditory Feedback	An initial postural measurement was performed. Biofeedback occurred when the angle of inclination exceeded ±10° of the sagittal plane and ±5° coronal plane from this position.	All subjects used the system for 2 h per day in this study. The trunk angles between 3-day trials were calculated.	A posture-monitoring system was used to estimate spinal curvature changes during trunk movements during daily activities.
Ribeiro et al., 2020 [60]	To evaluate the effectiveness of a lumbo-pelvic postural feedback device in changing postural behavior in a group of healthcare workers.	*n* = 130; age: 45.3 ± 13.2 Healthcare workers (Sham *n* = 67; Feedback *n* = 63)	Workers, with or without the presence (or history) of LBP.	D: Spine angel L: Hip	Auditory Intermittent Feedback	Feedback occurred when three signals (ROM, frequency, and duration) were activated. ROM: 45° lumbo-pelvic inclination forward. Frequency: when more than twice this threshold was exceeded for one minute. Duration: when inclination was maintained for 5 s.	Four weeks of intervention were conducted, during which participants wore the device in the workplace. On the first day of the intervention, they were informed that each time the biofeedback (sound) was given, they had to modify their posture until the beeping stopped.	Researchers counted the number of times the threshold was exceeded.
O’Sullivan et al., 2013 [65]	To study whether the use of postural biofeedback can reduce LBDs during a standardized sitting task.	*n* = 24; age: 24.7 ± 8.4	NSCLBP for at least 3 months.	D: BodyGuardTM L: L3 and S2	Vibration Feedback	A predefined threshold was set for each participant. If participants exceeded the threshold in that position, they received biofeedback to correct the position.	Sitting on a backless stool for 2 h.	

LBP: low back pain; CG: control group; IFG: intermittent feedback group (who received postural audio feedback in an alternating mode (1 week on, 1 week off) for 4 weeks. The postural audio feedback was inactive during the second and fourth weeks, with the lumbopelvic motion monitor set only to monitor over all 4 weeks of the intervention period; CFG: constant feedback group (who received postural audio feedback whenever the cumulative postural threshold was exceeded); h/w: hours per week; D: device; L: location; IG: intervention group; CLBP: chronic low back pain; NSCLBP: non-specific chronic low back pain.

## Data Availability

Data are contained within the article.

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
