# Peer review of "Lumbo-Pelvic Rhythm Monitoring Using Wearable Technology with Sensory Biofeedback: A Systematic Review"

_healthcare, 2024, doi:10.3390/healthcare12070758_

Round 1

Reviewer 1 Report

Comments and Suggestions for Authors

Dear Authors,

I appreciate the engaging topic you have chosen. However, I have a few concerns that I would like you to consider:

- Why were Embase, Cochrane Central, PsycINFO, CINAHL, and AMED omitted?

- Please consider citing https://www.mdpi.com/2073-8994/13/4/602 and  https://pubmed.ncbi.nlm.nih.gov/26566313/

- Before conducting a study, it is important to specify LBD as there are several differences between patients with a herniated disc, facet joint syndrome, spondylolisthesis, etc.

- The "sensors" section within the introduction needs improvement. Many devices were omitted.

- There is too little information provided about the participants' age and specific lower back pathology.

Thank you for taking the time to consider my feedback.

Comments on the Quality of English Language

Improvements needed. 

Author Response

Thursday, March 21, 2024

Dear Editor and Reviewers,

In light of the reviewers’ comments, we carefully revised our manuscript. We did our best addressing these remarks and we hope that the responses would be satisfactory to finally consider the revised manuscript for publication.

Please, find attached to the revised manuscript, now entitled “Lumbo-Pelvic Rhythm Monitoring Using Wearable Technology with Sensory Biofeedback: A Systematic Review”, (healthcare-2901664) a point-by-point response to the reviewers’ comments and questions. Finally, we would like to thank the reviewers for their constructive comments that have helped us to clarify several important aspects of this paper, improving its readability.

Kind regards,

Authors.

Reviewer 2 Report

Comments and Suggestions for Authors

Comments and Suggestions for Authors

Thank you for the opportunity to examine this interesting work that reviews wearable devices for LPR monitoring with reported biofeedback.

I consider it is necessary to improve the presentation of the study.

-It is recommended to register the review protocol in PROSPERO or in the Open Science Framework Registries (OSF) and provide the registration number.

KEYWORDS

-Put the first letter in capital letters in all cases. It is recommended that these terms conform as closely as possible to the controlled MeSH vocabulary to facilitate accurate document retrieval.

MATERIALS AND METHODS

-The search was limited from 2004 to 2024. For a systematic review to be as exhaustive as possible, it should not be limited by date, but rather should try to recover all existing information on the topic.

-Add as an inclusion criterion that the kinematic measurements were made with portable monitoring devices with a sensory biofeedback report and delete it from the exclusion criteria. It would also be pertinent to add the variables to be measured to the inclusion criteria. Reflect in the exclusion criteria the language if at any time it was a reason for exclusion, age, and study design.

-The STROBE scale (STrengthening the Reporting of Observational studies in Epidemiology) is specifically designed to evaluate the quality and risk of bias in observational studies (cohort studies, case-control studies, and cross-sectional studies). Therefore, it is not appropriate for clinical trials, case reports and preliminary studies. To evaluate the quality and risk of bias in these studies, the use of other specific tools is recommended, such as the CONSORT scale, PEDro, or the Cochrane Risk of Bias. It would be appropriate to make this modification.

-The Flowchart must be adjusted to the PRISMA format.

RESULTS

-Review the information that appears in LINES 178-181.

-The quality of the studies must be evaluated with the appropriate scales.

-The descriptive analysis of the studies should add the Study Population and the Variables to be measured in each case.

-The information between LINES 229-254 should be part of the Discussion.

-Table 3. Review header.

-“This systematic review is the first providing an overview of the wearable tracking devices”. This expression is repeated several times. It is advisable to delete it somewhere.

- 4.3: LINES 421-436 are not Practical applications.

CONCLUSIONS

-There are repeated Conclusions.

REFERENCES

-Review the format of this section.

-The title of the journals must appear in abbreviated format.

-Check the number of authors in all references. Normally, after the 6th author we put et al.

-It is recommended to add DOI to references that have it.

-REF Nº 14: it is incomplete.

-REF Nº 29: Review this reference by entering: Author(s) of the book. Title of the book. Edition number ed. Place of publication: Publisher; Year of publication. Number of pages. ISBN

-REF Nº 66: it is incomplete.

Author Response

(The authors gave the same response as above.)

Reviewer 3 Report

Comments and Suggestions for Authors

This study will conduct an important literature review on the mechanisms that may cause lumbar and lower back diseases.

1. The number of 8 papers finally selected is judged to be too small for a literature review. Please write down the author's thoughts.

2. The final conclusion appropriate to the topic is not clear. Please state again a clear conclusion that fits the research purpose.

Comments on the Quality of English Language

English expressions are not perfect.

Author Response

(The authors gave the same response as above.)

Round 2

Reviewer 1 Report

Comments and Suggestions for Authors

Dear Authors,

this manuscript has improved.

Comments on the Quality of English Language

Minor corrections needed.

Author Response

Sunday, March 24, 2024

Dear Editor and Reviewers,

In light of the reviewers’ comments, we carefully revised our manuscript. We did our best addressing these remarks and we hope that the responses would be satisfactory to finally consider the revised manuscript for publication.

Please, find attached to the revised manuscript, now entitled “Lumbo-Pelvic Rhythm Monitoring Using Wearable Technology with Sensory Biofeedback: A Systematic Review”, (healthcare-2901664) a point-by-point response to the reviewers’ comments and questions. Finally, we would like to thank the reviewers for their constructive comments that have helped us to clarify several important aspects of this paper, improving its readability.

Authors’ response: Thank you for your comment. Your valuable comments have helped us a lot to improve the paper.

Kind regards,

Authors.

Reviewer 2 Report

Comments and Suggestions for Authors

I agree with most of the answers provided by the authors, however there are some aspects that still need to be improved in the manuscript.

-The STROBE scale is used as a check list for observational studies (cohort studies, case-control studies and cross-sectional studies). Indeed in the study “O’Reilly, M.; Caulfield, B.; Ward, T.; Johnston, W.; Doherty, C. Wearable inertial sensor systems for lower limb exercise detection and evaluation: A systematic review. Sports Med. 2018, 48, 1221–1246”, the quality of included studies was analyzed using an adapted version of the STROBE assessment criteria for cross-sectional studies, which is an observational study modality. However, it is not an appropriate scale to evaluate the methodological quality of other studies such as clinical trials, randomized controlled trials, care reports or preliminary studies. It is recommended to use the appropriate scales.

-On the other hand, it is recommended to review the PRISMA Flow chart conform to the 2020 format. PRISMA (prisma-statement.org)

Author Response

Sunday, March 24, 2024

Dear Editor and Reviewers,

In light of the reviewers’ comments, we carefully revised our manuscript. We did our best addressing these remarks and we hope that the responses would be satisfactory to finally consider the revised manuscript for publication.

Please, find attached to the revised manuscript, now entitled “Lumbo-Pelvic Rhythm Monitoring Using Wearable Technology with Sensory Biofeedback: A Systematic Review”, (healthcare-2901664) a point-by-point response to the reviewers’ comments and questions. Finally, we would like to thank the reviewers for their constructive comments that have helped us to clarify several important aspects of this paper, improving its readability.

Kind regards,

Authors.

Round 3

Reviewer 2 Report

Comments and Suggestions for Authors

I am sorry that the previous revision has not been interpreted correctly. I consider it necessary to introduce the following comments in order to improve the manuscript:

There is one descriptive study in the sample of the 8 studies analysed. The PEDro scale is not suitable for directly assessing the methodological quality of descriptive studies. For this specific case, it is recommended to use the STROBE scale, as was done initially.

Review in SCREENING of the Flow chart PRISMA the difference between the 222 Reports assessed for eligibility and the Reports excluded n= 185. The result would be 37 studies included in review +2. Cells n=0 can be deleted.

Author Response

Tuesday, March 26, 2024

Dear Editor and Reviewers,

In light of the reviewers’ comments, we carefully revised our manuscript. We did our best addressing these remarks and we hope that the responses would be satisfactory to finally consider the revised manuscript for publication.

Please, find attached to the revised manuscript, now entitled “Lumbo-Pelvic Rhythm Monitoring Using Wearable Technology with Sensory Biofeedback: A Systematic Review”, (healthcare-2901664) a point-by-point response to the reviewers’ comments and questions. Finally, we would like to thank the reviewers for their constructive comments that have helped us to clarify several important aspects of this paper, improving its readability.

Kind regards,

Authors.
